# Development and Implementation of a Single Radial Diffusion Technique for Quality Control of Acellular Pertussis Vaccines

**DOI:** 10.3390/vaccines13020116

**Published:** 2025-01-24

**Authors:** Chongyang Wu, Xi Wang, Yu Zhou, Xinshuo Zhu, Yu Ma, Wenming Wei, Yuntao Zhang

**Affiliations:** 1Department of Research and Development, Beijing Institute of Biological Products Co., Ltd., Beijing 100176, China; wuchongyang@sinopharm.com (C.W.); wangqian165@sinopharm.com (X.W.); 2Bacterial Vaccine Manufacturing Facility, Beijing Institute of Biological Products Co., Ltd., Beijing 100176, China; 3China National Biotec Group Co., Ltd., Beijing 100024, China

**Keywords:** acellular pertussis vaccine components, antigen content quantification, single radial immunodiffusion, analytical methods validation, quality control

## Abstract

Background/Objectives: An assay for protein content is essential but insufficient for quality control of acellular pertussis vaccines, which might consist of up to five components, each needing individual quantification. Generally, purified pertussis antigens such as pertussis toxin (PTx), filamentous haemagglutinin (FHA), and pertactin (PRN) should be detoxified or stabilized chemically before being formulated into vaccine bulk. The use of chemical agents like formaldehyde and glutaraldehyde can alter the immunological reactivity of these antigens, rendering direct assays by methods such as ELISA ineffective. Methods: In this study, a simple method based on single radial diffusion (SRD) using low concentrations of polyclonal antisera against PT toxoid (PTd), FHA, and PRN was developed. By adding a detergent, diffusible subunits are produced regardless of the original physical state of the antigens, making it suitable for quantifying these antigens after chemical treatment. Results: The assay has shown good specificity, accuracy, and precision. Furthermore, it can differentiate between preparations with the same protein concentration but different antigenic contents. A significant positive correlation between the antigen content and the in vivo immunogenicity has also been demonstrated. Conclusions: An assay for quality control and consistency monitoring of combined vaccines containing acellular pertussis antigen components has been established.

## 1. Introduction

Pertussis, also known as whooping cough, remains one of the most common contagious diseases causing morbidity and mortality worldwide, despite the high vaccination rate with two types of vaccine—whole-cell pertussis vaccine (WPV) and acellular pertussis vaccine (APV) [1,2,3]. Neither WPV nor APV provides lifelong protection in infants or young children, so booster immunizations are necessary for teenagers and adolescents [4,5]. Due to safety concerns of WPV, APV is preferred despite the rapid decrease in antibody levels following vaccination with APV [6,7]. Most European countries and some Asian (Japan, Korea, Australia, and China), African, and Latin American countries, have incorporated APV into their immunization schedules [8].

APV consists of up to five antigen components, including pertussis toxoid (PTd), filamentous haemagglutinin (FHA), pertactin (PRN), and fimbrial agglutinogens (Fim 2/3). However, the specific composition can vary significantly between manufacturers, and the components may be prepared by co-purifying or separate purification of the harvested material. The exact amount of each antigen is measured only in the case of separate purification, before being formulated into vaccine bulks. However, it is crucial to assay each component separately in the manufacturing process of APV to ensure quality.

Protein content quantification is commonly performed for assaying antigen components, which is essential but not sufficient, particularly for antigens treated with detoxifying or stabilizing chemical agents like glutaraldehyde or formaldehyde. These agents always lead to the formation of high molecular weight components [9,10], which can change the immunological reactivity of the native antigen due to the impacted presentation of neutralizing epitopes [11,12]. In such cases, an additional assay of antigen content is necessary as protein content alone may not accurately represent the antigen’s immunogenicity. Although ELISA can effectively quantify purified antigens [13], it is not suitable for detoxified antigens due to the modification effects caused by chemical agents.

An APV developed by local Chinese manufacturers using a separate purification process is currently in the clinical research stage. Although the total protein assay is utilized to monitor individual components after detoxification, it may not accurately reflect the true antigenic content post-detoxification. Given the limitations of direct ELISA, it is important to develop an alternative assay to accurately quantify antigen content, independent of the impact of aldehyde reagents.

The single radial immunodiffusion (SRD) technique, an efficient method of determining antigen content, can measure the total content of the antigen, thereby bypassing the impacts caused by detoxifying agents. This technique, initially described by Schild et al. to assay influenza hemagglutinin (HA) antigen [14], has proven its efficiency over the years. Its accuracy and productivity have been verified, making it a potent assay for subunit influenza vaccines [15]. This work established and validated a simple and efficient SRD using polyclonal antibodies suitable for assessing the antigen content of PTx, FHA, and PRN, underscoring the efficiency and accuracy of the SRD technique.

## 2. Materials and Methods

### 2.1. Reagents and Buffers

Zwittergent 3-14 detergent (code: 693017) was purchased from Millipore (Burlington VT, USA), agarose (code: BY-R0100) was from BIOWEST (Nuaillé, France), Freund’s adjuvant (code: P5881) and glutaraldehyde (code: G6257) were purchased from Sigma (Tokyo, Japan), Coomassie brilliant blue (code: 6104-59-2), methanol, and acetic acid were from Sinopharm Chemical Reagent Co., Ltd. (Shanghai, China). The lysis buffer (PBS containing 1% Zwittergent), staining solution (0.2% (*w*/*v*) Coomassie brilliant blue, 5% (*v*/*v*) methanol, 1% (*v*/*v*) acetic acid in distilled water), and destaining solution (40% (*v*/*v*) methanol, 10% (*v*/*v*) acetic acid in distilled water) were prepared in the lab.

### 2.2. Antigens and Vaccine Candidates

Pertussis antigens PTx, FHA, PRN and detoxified preparations were manufactured by the bacterial vaccine manufacturing facility of the Beijing Institute of Biological Products Co., Ltd. (BIBP) (Beijing, China). Acellular pertussis vaccine candidates were prepared in the lab.

### 2.3. Animals and Antisera

The big-eared white rabbits, 4 months of age and about 3 kg, were obtained from Beijing Regal Laboratory Animal Breeding Center (Beijing, China). The NIH mice—female, 5–6 weeks of age, and 11–13 g—were obtained from SPF (Beijing) Biotechnology Co., Ltd. (Beijing, China). All animals were bred and maintained in a specific pathogen-free (SPF) environment. The immune sera to detoxified PTx, FHA, and PRN were prepared in the lab. First, Freund’s complete adjuvant was used for sensitization, and 0.3 mL (0.6 mL in total) was injected into each rabbit’s foot. Two weeks later, 500 μg/mL antigen was added into the Freund’s complete adjuvant, mixed in 1:1 volume, fully emulsified, and 0.4 mL was injected into the thigh muscle; a total of 0.8 mL. Three weeks later, 500 μg/mL antigen was added into Freund’s incomplete adjuvant, mixed in 1:1 volume, fully emulsified, and 1 mL was injected subcutaneously on the back. After that, multiple injections were given to the back subcutaneously once every two weeks to enhance immunization 3 times. Blood was collected 14 days after the third boost immunization, and immune sera were prepared by centrifugation.

All animals involved in this study were housed and cared for in an Association for the Assessment and Accreditation of Laboratory Animal Care (AAALAC)-accredited facility. All experimental procedures were conducted according to Chinese animal use guidelines and were approved by the Institutional Animal Care and Use Committee (IACUC). All animals were anesthetized using isoflurane.

### 2.4. ELISA Assays for Estimating Antibodies to PTx, FHA, and PRN

Antibodies against PTx, FHA, and PRN were determined by ELISA assays as previously described [16,17]. Briefly, 96-well microtitration plates (Greiner Bio-one, Frickenhausen, Germany) were pre-added with the purified PTx, FHA, and PRN antigens and kept overnight at 4 °C. The plates were washed 4 times with PBS containing 0.05% Tween 20 (PBS-T20, washing buffer). For blocking, 100 μL of PBS containing 1% bovine serum albumin (assay buffer) was added and incubated at 37 °C for one hour. After the assay buffer was aspirated, NIBSC reference serum (eight steps of 2-fold dilutions (1/100 to 1/218,700) were prepared with assay buffer) and immune sera were added (diluted 2-fold for seven dilutions starting at 1/100), followed by the addition of secondary peroxidase-labeled anti-mouse antibodies (Sera Care, Milford, MA, USA). The absorbance of each well was read using a Multiskan FC reader (Thermo Fisher Scientific, Waltham, MA, USA) at 450/630 nm. The PTx, FHA, and PRN antibody levels were expressed in IU/mL.

### 2.5. Assay Development

#### 2.5.1. Preparation of SRD Plates

The SRD plates were prepared as described [15]. The agarose solution was cooled down in a 50 °C water bath, and rabbit antisera were added into the agarose at final concentrations of 6.67 μL/mL for PTx, 4 μL/mL for FHA, and 5 μL/mL for PRN. Following this, agarose gel plates were prepared by pouring the agarose-sera solution into dishes (12 × 8 cm^2^). Then, sample wells were made using a 3 mm diameter metal cutter.

#### 2.5.2. Performance of SRD Assays

The reference and antigen samples were diluted in PBS containing 1% (*w*/*v*) Zwittergent detergent (lysis buffer) to an initial protein concentration of 200 μg/mL or 400 μg/mL. The antigen solutions were incubated at room temperature for a minimum of 30 min. Subsequently, several steps of 2-fold dilutions were prepared in lysis buffer prior to loading into the wells. Volumes of 20 μL (2 × 10 μL) of the appropriate dilution of samples and the reference were introduced into the wells in the agarose.

#### 2.5.3. Processing and Reading of SRD Plates

The gel plate was positioned horizontally in a humidified chamber and incubated at 37 °C for 24 h. After that, it was rinsed with PBS and purified water and subsequently dyed for 5~10 min before undergoing further washing steps. The gel was destained two to three times for 10–15 min each until diffusion zones were visible and then rinsed with purified water. The gel was scanned and photographed, and the diameters of diffusion zones were measured using the Immulab Image Analysis Software (MICROVISION Instruments SAS, CE 1750—Z.I. Petite Montagne Sud 1 rue du Gévaudan 91,047 EVRY Cedex—France, file version:12.7.0.1). Dose–response curves of antigen dilution against the zone annulus area were constructed, and the results were calculated according to standard parallel line assay methods. If necessary, the relative antigen content is calculated as antigen content divided by protein concentration.

### 2.6. Assay Validation

#### 2.6.1. Working Range and Detection Limit

Purified PTx (in-house reference) and detoxified PTx (test sample) were incubated with lysis buffer at room temperature for 30 min. Subsequently, five-steps 2-fold dilutions were prepared, ranging from 200 to 6.25 μg/mL. To confirm the linear range of the assay, dose–response curves of antigen dilution against the zone annulus area were plotted using standard parallel line assay methods. The coefficient of determination and regression significance of the linear equation derived from the selected concentration range (dilutions range) were thoroughly analyzed. Additionally, the linear range and parallelism were further confirmed multiple times with the selected concentration range. Similarly, the linear range of the assay for FHA and PRN was carried out.

#### 2.6.2. Specificity

To validate the assay’s specificity for PTx, a batch of purified PTx antigen was used as positive control, a batch of purified FHA and PRN antigen was used as interference control, and lysis buffer was treated as negative control. The samples were loaded as mentioned above in the same arose gel plate containing antisera to PTd, and diffusion zones were observed 24 h later. Acceptance criteria: no diffusion zones are produced around the wells of negative control and interference control, and clear diffusion zones are formed around the wells of positive control. Similarly, the specificity of the assays for FHA and PRN was validated.

#### 2.6.3. Accuracy

To validate the accuracy of the assay, three spiked samples containing different amounts of antigens were independently prepared for PTx, FHA, and PRN. The theoretical antigen content was calculated before the samples were tested. Each spiked sample was tested three times, and the measured values of antigen content were averaged. The samples’ recovery rate and relative standard deviation (RSD) were then meticulously calculated based on the measured and theoretical values. Acceptance criteria: the recovery rate of the sample should be between 80% and 120%, and the RSD should not be more than 15%.

#### 2.6.4. Precision

To validate the precision of the assay, the reproducibility of the assay was first investigated; one batch of detoxified PTx, FHA, and PRN was individually analyzed for relative antigen content by the first experimenter six consecutive times. The RSD_6_ was determined based on six results. Acceptance criterion: the RSD_6_ should not exceed 15%. In the context of intermediate precision, the identical batch of detoxified PTx, FHA, and PRN samples were subject to testing by the second experimenter for another six consecutive times. Both RSD_6_ and RSD_12_ (RSD of 12 results) were calculated. Acceptance criterion: the RSDs should not exceed 15%.

### 2.7. Application of the Assay

#### 2.7.1. Correlation Analysis of Antigen Content and Antibodies Response

Acellular Pertussis vaccines containing different amounts of PTd (100, 50, 25 μg/mL), FHA (100, 50, 25 μg/mL), and PRN (32, 16, 8 μg/mL) antigens were prepared at high, medium, and low doses. NIH mice were given 0.5 mL of 1/3 HSD (human single dose) vaccine samples intraperitoneally on day 0 and day 21. On day 35, serum samples were collected via orbital bleeding, and serum antibodies to PTx, FHA, and PRN were determined by ELISA assays, as mentioned in Section 2.4. The results were then plotted based on antigen content and antibody response, and Pearson correlation was analyzed.

#### 2.7.2. Assay of Antigen Content of PTd, FHA, and PRN Bulk Materials

First, the SRD developed in the lab was used to assay PTd samples with different detoxification times. Furthermore, 23 batches of detoxified PTx antigens, 15 batches of FHA antigens, and 13 batches of PRN antigens were quantified using SRD to assess the method’s applicability.

## 3. Results

### 3.1. Optimization and Standlization of SRD Procedures

While developing the SRD method, we studied the impact of immune serum concentration, agarose concentration, lysate concentration, and lysis duration on the results. Our findings for the assay for FHA are as follows. (1) Serum dilutions of 150×, 200×, and 250× showed no significant difference in area and clearness of diffusion zones when samples were loaded at the same concentration (Figure 1A). Therefore, we selected 250 times dilution as the optimal antiserum concentration due to its efficiency in using less sera. (2) There was no significant difference in diffusion zones between 1% and 1.5% agarose gel (Figure 1B). However, the 1.5% agarose gel provided better flexibility and integrity during staining and destaining. As a result, we chose the 1.5% agarose gel. (3) Testing the effect of lysate concentration revealed no significant difference in diffusion zones for samples treated with PBS containing 1%, 2%, and 5% Zwittergent, respectively (Figure 1C). We chose the PBS containing 1% Zwittergent as the lysis buffer for cost-saving reasons. (4) Extending the lysis duration from 0.5 h to 1 h and 2 h did not result in any significant change in diffusion zones (Figure 1D). To improve efficiency, we selected a lysis duration of 0.5 h. After establishing these parameters, we developed the assays for PTd, FHA, and PRN. Finally, all the parameters were determined for the SRD method specific to these three antigen components (see Table 1).

### 3.2. Validation of SRD

#### 3.2.1. Linear Range Confirmation

The dose–response linear regression lines of purified PTx (in-house reference) and detoxified PTx (test sample) at the concentration range from 25 μg/mL to 200 μg/mL demonstrated that the coefficient of determination and regression significance of the linear equation were the best overall. At the same time, the reaction zone (precipitated ring) was not clear when the antigen concentration was lower than 12.5 μg/mL. Accordingly, three dilutions of 25, 50, and 100 μg/mL were selected to re-confirm the linear range and parallelism through multiple independent experiments. As shown in Figure 2A–D, the in-house reference and test material exhibited highly significant (regression factor, *p* < 0.01) dose–response curves, and the deviations from parallelism and linearity were not significant (*p* > 0.05). Similar work was carried out for FHA and PRN in-house reference, the concentration ranges were determined as 12.5 μg/mL~200 μg/mL and 12.5 μg/mL~50 μg/mL for them, respectively.

#### 3.2.2. Validation on the Specificity

Specificity validation of the SRD was accomplished according to the standard procedure and linear range mentioned in Section 2.1 and Section 2.2. As depicted in Figure 3A, reaction zones with distinct areas (diameters) were observed at various concentrations of PTx reference. However, there were no reaction zones around the well of the other two pertussis components (FHA and PRN) and the lysis buffer (negative control). Similar results were observed for FHA and PRN references; antisera to each specifically reacted with FHA or PRN. There was no cross-reaction among these three pertussis components, proving that the SRD technique specifically identifies PTx, FHA, and PRN.

#### 3.2.3. Validation on the Accuracy

In this section, three spiked samples of each pertussis component were used to validate the accuracy. As shown in Table 2, the recovery of antigen content of all the PTx, FHA, and PRN spiked samples ranged from 88.9% to 118.5%, 97% to 113%, and 99.7% to 104.9%, respectively. The RSDs of the results ranged from 9.6% to 10.7%, 2.5% to 4.1%, and 2.1% to 6.0%. All of these results meet the acceptable criteria and indicate good accuracy.

#### 3.2.4. Validation on the Precision

Firstly, to investigate the method’s repeatability, one batch of detoxified PTx, FHA, and PRN was individually analyzed for relative antigen content by the first experimenter six consecutive times. The RSD_6_ was determined based on six results. Subsequently, intermediate precision was evaluated as follows. The identical batch of detoxified PTx, FHA, and PRN samples was subject to testing by the second experimenter for another six consecutive times. Both RSD_6_ and RSD_12_ (RSD of 12 results) were calculated. The RSDs depicted in Table 3 all fell below 15%. These results demonstrate the precision of the method taken together.

### 3.3. Correlation of Antigen Content and In Vivo Immunogenicity

Correlations of antigen content and in vivo immunogenicity are shown in Figure 4. The correlation coefficient (Pearson’s r) of each is 0.62 (Figure 4A), 0.65 (Figure 4B), and 0.61 (Figure 4C), indicating a positive correlation between antigen content and antibody responses in vivo for PTd, FHA, and PRN.

### 3.4. Monitoring the Consistency of PTd, FHA, and PRN Bulk Materials

Using the established SRD method to analyze the effect of different detoxification durations (10 min to 24 h) on detoxified PTx antigens, the results indicated that the antigen content of detoxified PTx tended to decrease with increasing detoxification time. (Figure 5A). Furthermore, the method’s applicability was assessed by continuously monitoring the antigen content of multiple consistent detoxified PTx, PRN, and FHA batches. The values for 23 batches of detoxified PTx, 13 batches of PRN, and 15 batches of FHA were found to be within $\bar{x}$ ± 3SD, with the values evenly distributed on either side of the mean value (see Figure 5B–D). In conclusion, the method is well-suited for detecting antigen content in detoxified PTx, FHA, and PRN intermediates.

## 4. Discussion

Pertussis is a contagious respiratory disease that causes morbidity and mortality in children under five years old and has had a resurgence in recent years, with increasing incidence even in countries with high vaccine coverage [2,18,19,20]. Currently, vaccination with APV or WPV series combined vaccines (referring to DTaP and DTwP) is a primary method to prevent the disease. Due to safety concerns, DTaP is a preferred vaccine in industrialized countries. Monitoring the consistency of pertussis antigen components is vital to ensure the quality of acellular pertussis vaccines. European Pharmacopoeia version 9.0 (published in 2013) and the WHO guideline in the Technical Report Series (TRS) 979 Appendix 4 clearly state that the antigen content of purified pertussis components should be determined before being formulated in final bulks. Furthermore, the ratio of antigen content to protein nitrogen should be within the established limits. In other words, this requirement ensures that acellular pertussis-derived combined vaccine products marketed in Europe and countries undergoing WHO pre-qualification programs must strictly monitor any changes in antigen properties and quantify antigen content post-chemical-detoxification or fixation.

The commonly available DTaP vaccines in China are produced by a co-purification process. While most manufacturers use protein content to monitor the consistency of pertussis antigen components after detoxification, this method can only reflect the total protein amount but not characterize the changes in antigenic properties. Notably, treating pertussis antigen components with formaldehyde or glutaraldehyde could lead to the loss of some crucial epitopes that were proven to confer protection in vivo [21,22]. Our unpublished work found that neutralizing epitopes on the S1 (subunit 1) and S2/3 (subunit 2/3) of PTx decreased by at least 30% after treatment with 0.05% glutaraldehyde for 4 h. Increasing the amount of glutaraldehyde or extending the incubation time can cause further loss of neutralizing epitopes, potentially exceeding 80%, with only a mild impact on protein content. Therefore, developing a technique to assay the antigen content is necessary and urgent to guide the development and production of DTaP vaccines of high quality in China.

Glutaraldehyde, a chemical agent commonly used to treat pertussis antigens, such as PTx, FHA, and PRN, often leads to the formation of high molecular weight by-products due to its crosslinking effects. This crosslinking can obscure critical binding sites on the surface of the antigens, rendering the ELISA assay ineffective. This limitation of the ELISA assay is an important consideration in the field. In contrast, the SRD assay can accurately measure the total antigen content, regardless of any changes to the binding sites. This is accomplished using a detergent, which enables the production of diffusible subunits, irrespective of their original state.

The assay described in this paper was based on the work of Dr. Xing and colleagues [23], who demonstrated that the assay could serve as an in-process control method for acellular pertussis vaccines. However, they did not address the specificity, accuracy, precision, and robustness concerning the validation of analytical methods. Therefore, we conducted a systematic method validation as a supplementary measure. First, we demonstrated its reasonable specificity without cross-reactivity between the three components. The assay also exhibits good accuracy, with a recovery rate of spiked samples ranging from 85% to 120%. Additionally, the relative standard deviation (RSD) for inter- and intra-variability is less than 15%, proving a good precision.

Furthermore, variations in antisera concentration, incubation time, detergent concentration, and agarose concentration have a mild impact on the results, indicating good robustness.

It is important to note that the theoretical values of the selected spiked samples did not cover the lower limit of the detection range. In other words, more thorough validation of accuracy and precision should be conducted in the future. Notably, the quality of rabbit sera varies between batches, and this variation’s effect on the assay’s robustness should be investigated in future work. In addition, according to Dr. Xing and colleagues’ work, the assay was unsuitable for the other two components, Fim2 and Fim3. Since these components were not included in our vaccine formulations, we did not test Fim2 and Fim3 in the SRD system.

After its establishment, the SRD assay was utilized to assess PTd samples with varying levels of detoxification. This assay is sensitive enough to detect differences in antigen content among different preparations that share the same protein content. Furthermore, the antigen content of consistent batches of detoxified PTx, FHA, and PRN was monitored using SRD. All batches of these bulk preparations demonstrated antigen content within $\bar{x}$ ± 3SD, indicating good batch-to-batch consistency. The study also revealed a positive correlation between the antigen content of PTd, FHA, and PRN and their immunogenicity in vivo. This implies that antigen content could potentially be used to guide vaccine formulation. It is important to note that this method measures antigen content directly rather than antigenicity or immunogenicity. Although a strong correlation between antigen content and immunogenicity was observed in animal experiments, the direct relationship with protection or effectiveness cannot be established based solely on in vitro antigen content detection. This is due to the influence of numerous factors, including the subject’s physiological status, age, and vaccination history with similar antigens, on the antibody response in vivo.

## 5. Conclusions

In conclusion, this study established an SRD technique that can be used to assay pertussis antigens, including PTx, FHA, and PRN, post-detoxification. This method has successfully monitored the consistency between batches of detoxified preparations. Additionally, it could replace the protein content quantification method currently used to guide vaccine formulation.

## Figures and Tables

**Figure 1 vaccines-13-00116-f001:**
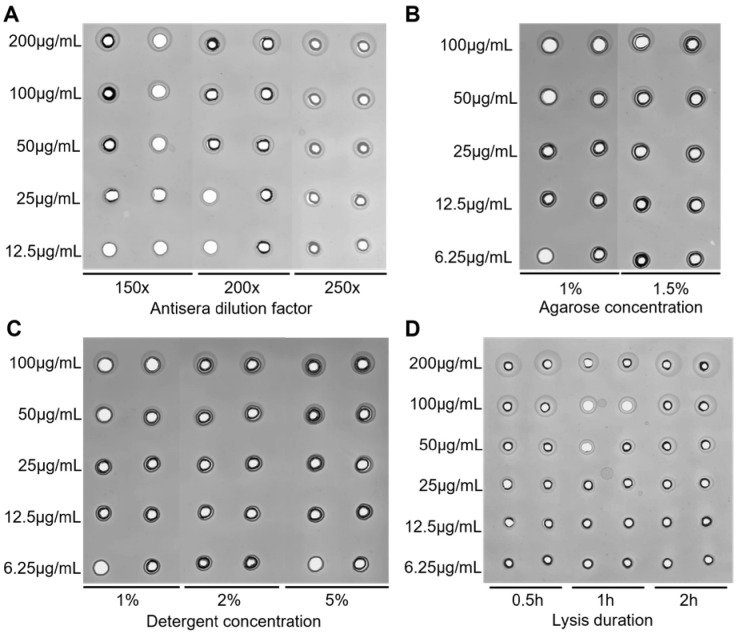
Optimization of experiment parameters for SRD. (**A**) Examination of antisera concentration in gel; FHA samples with different concentrations diffused in gels containing 150×, 200×, and 250× diluted antisera were studied. (**B**) Determination of agarose concentration in gel; FHA samples with different concentrations diffused in gels containing 1% and 1.5% agarose were investigated. (**C**) Investigation of detergent concentration; FHA samples were incubated with assay buffers containing different concentrations (1%, 2%, and 5%) of detergent for at least 30 min. (**D**) Examination of lysis duration; FHA samples with various concentrations were incubated with assay buffer for 0.5 h, 1 h, and 2 h.

**Figure 2 vaccines-13-00116-f002:**
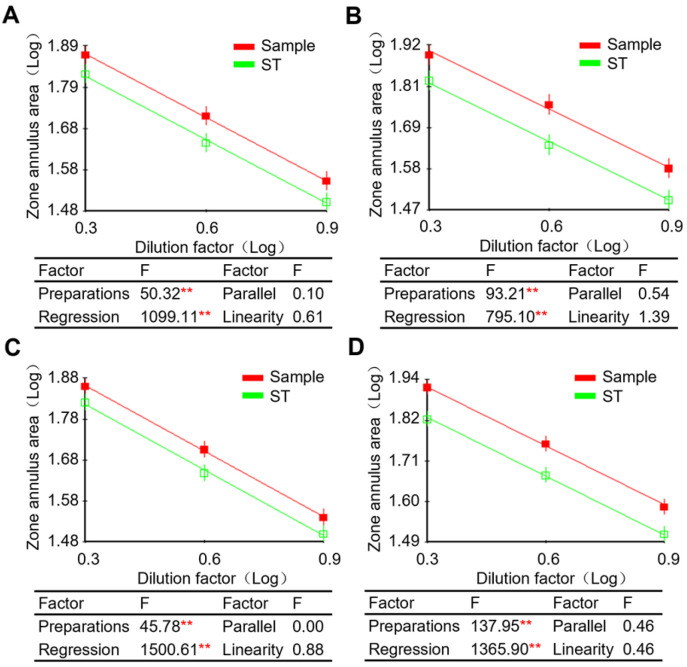
Determination of working range and detection limit. Both in-house reference and test samples were loaded with concentrations ranging from 100 to 25 μg/mL, and dose–response curves were plotted on antigen dilution factor (Log) against zone annulus area (Log) using standard parallel line assay methods. The coefficient of determination and regression significance of the linear equation derived from the selected concentration range were analyzed. The linear range and parallelism determination experiment was performed multiple times. (**A**) Experiment# 1; (**B**) Experiment# 2; (**C**) Experiment# 3; (**D**) Experiment# 4. ** 1% Significance level.

**Figure 3 vaccines-13-00116-f003:**
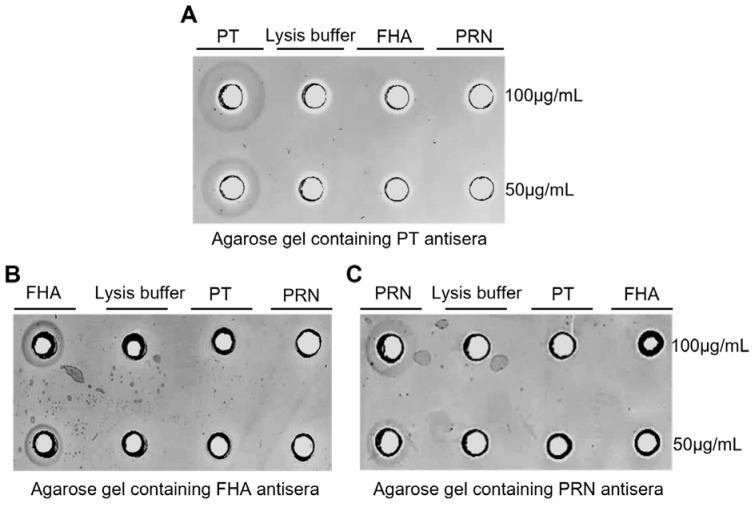
Validation on specificity of SRD assays. (**A**) Two samples for each of purified PTx, FHA, PRN and lysis buffer were loaded in gel containing antisera against PTd; (**B**) purified FHA, PTx, PRN and lysis buffer were loaded in gel containing antisera against FHA; and (**C**) purified PRN, PTx, FHA and lysis buffer were loaded in gel containing antisera against PRN.

**Figure 4 vaccines-13-00116-f004:**
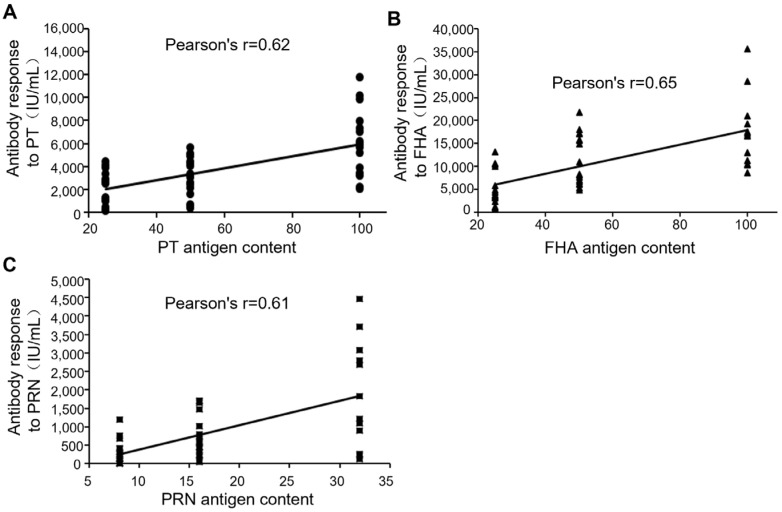
Correlation of antigen content and antibody response in mice immunized with corresponding vaccine formulations containing PTd, FHA, and PRN. The antibody response (geometric mean titer) was measured by ELISA as described in Section 2. (**A**) PTd. (**B**) FHA. (**C**) PRN. Filled circles: immune sera samples raised by vaccine formulations containing various content of PTd; Filled triangles: immune sera samples raised by vaccine formulations containing various content of FHA; Filled squares: immune sera samples raised by vaccine formulations containing various content of PRN.

**Figure 5 vaccines-13-00116-f005:**
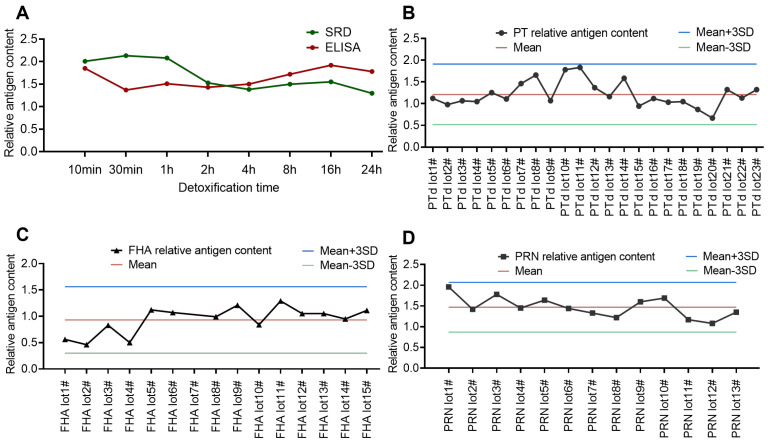
Application of SRD in quality control testing of PTd, FHA, and PRN bulk materials. (**A**) The antigen content of PTd preparations of different detoxification times were analyzed by SRD. (**B**) SRD is used for monitoring consistency of 23 batches of PTd bulk materials. (**C**) SRD is used for monitoring consistency of 15 batches of FHA bulk materials. (**D**) SRD is used for monitoring consistency of 13 batches of PRN bulk materials.

**Table 1 vaccines-13-00116-t001:** Parameters determined for SRD assays specific to three pertussis components.

Items	PTd	FHA	PRN
lysis buffer	PBS containing 1% (*w*/*v*) Zwittergent	PBS containing 1% (*w*/*v*) Zwittergent	PBS containing 1% (*w*/*v*) Zwittergent
lysis temperature/duration	room temperature, 30 min	room temperature, 30 min	room temperature, 30 min
incubation temperature/duration	37 °C, 24–28 h	37 °C, 24–28 h	37 °C, 24–28 h
antisera dilution factor	1/150	1/250	1/200
agarose concentration	1.5%	1.5%	1.5%

**Table 2 vaccines-13-00116-t002:** Results on validation of accuracy.

Spiked Samples	Relative Antigen Content (Theoretical Value)	Relative Antigen Content (Measured Value)			Recovery	RSD
		Test 1	Test 2	Test 3		
Spiked PTd 1#	0.913	0.763	0.815	0.92	91.20%	9.60%
Spiked PTd 2#	1.038	0.898	0.838	1.031	88.90%	10.70%
Spiked PTd 3#	1.163	1.404	1.421	1.378	118.50%	4.30%
Spiked FHA 1#	0.845	1.005	0.958	0.897	113%	4.00%
Spiked FHA 2#	0.97	1.067	1.033	0.993	106%	2.50%
Spiked FHA 3#	1.095	1.128	1.065	0.998	97%	4.10%
Spiked PRN 1#	0.635	0.579	0.631	0.69	99.70%	6.00%
Spiked PRN 2#	0.76	0.779	0.791	0.823	104.90%	2.10%
Spiked PRN 3#	0.885	0.872	0.914	0.918	101.80%	2.20%

**Table 3 vaccines-13-00116-t003:** Results on validation of precision.

Samples	Experimenter	Relative Antigen Content							
		Test 1	Test 2	Test 3	Test 4	Test 5	Test 6	RSD_6_	RSD_12_
PTd	1	0.834	0.864	0.914	0.881	0.803	0.746	7.10%	10.90%
	2	0.865	0.699	0.893	0.624	0.845	0.75	13.60%	
FHA	1	0.961	0.812	0.925	0.958	0.822	0.851	6.73%	6.58%
	2	0.975	0.931	0.958	0.775	0.915	0.968	5.48%	
PRN	1	0.664	0.822	0.715	0.66	0.685	0.651	13.20%	12.50%
	2	0.865	0.699	0.893	0.624	0.845	0.75	13.30%	

## Data Availability

The data presented in this study are available on request from the corresponding author. The data are not publicly available due to restrictions.

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
