# Peer review of "Development and Implementation of a Single Radial Diffusion Technique for Quality Control of Acellular Pertussis Vaccines"

_vaccines, 2025, doi:10.3390/vaccines13020116_

Round 1
Reviewer 1 Report
Comments and Suggestions for Authors
The authors use a single radial diffusion (SRD) technique to assess antigen content of inactivated acellular pertussis vaccine components. The approach was previously developed. The authors reasonably rationalize its advantages over quantification by protein content and ELISA reactivity, and its potentially utility for standardization of commercial vaccine preparations. The experimental design is straightforward and appears to have been performed rigorously with testing of different parameters and inclusion of appropriate controls. The results are clear and the interpretations and conclusions are sound. Still, the technical complexities of the SRD approach should be acknowledged, although this point does not detract from the value of this manuscript. For instance, polyclonal antisera must be raised in rabbits and used to prepare SRD agarose plates, followed by use of specialized software to analyze diffusion results. Despite the limitations of other easier and faster quantitation techniques (protein content, ELISA reactivity) relative to SRD, the greater technical complexities of SRD may provide an impediment to practical implementation.
My only specific comment relates to Materials & Methods lines 124-126. What was the basis for determining the final concentrations of rabbit antisera for the 3 different antigens in preparing the SRD agarose plates?
Author Response
Comments 1: My only specific comment relates to Materials & Methods lines 124-126. What was the basis for determining the final concentrations of rabbit antisera for the 3 different antigens in preparing the SRD agarose plates?
Response 1: Thank you for reviewing our manuscript; I appreciate your valuable comment. To clarify, we tested three different concentrations of rabbit antisera for each of the three antigens. In determining the appropriate concentrations, we considered two criteria: first, the diffusion zones needed to be large enough and identifiable; second, we aimed to minimize costs using the smallest volume of sera possible.
Reviewer 2 Report
Comments and Suggestions for Authors
Pertussis is one of the most common contagious diseases, causing mobility and mortality in infants and young children less than 5 years worldwide, despite the high vaccination rate with two types of vaccine - whole cell pertussis vaccine (WPV)and acellular pertussis vaccine (APV). APV consists of antigens, including pertussis toxoid (PTd), filamentous haemagglutinin (FHA), pertactin (PRN), and fimbrial agglutinogens (Fim). However, the specific composition can vary significantly between manufacturers. Therefore, it is crucial to assay each component separately in the manufacturing process of APV for quality control purposes. Protein content quantification is a standard assay applied for assaying antigen components, which is essential but insufficient, particularly for antigens treated with detoxifying or stabilizing chemical agents like glutaraldehyde or formaldehyde. It is necessary to develop an alternative assay to accurately quantify antigen content, independent of the impact of aldehyde reagents. This manuscript reported the development of a single radial immunodiffusion (SRD) technique that can determine antigen content regardless of the physical state of the antigen but the total content, which circumvents the impacts caused by detoxifying agents. The authors established and validated a relatively simple and efficient SRD using polyclonal antibodies suitable for assessing the antigen content of PT, FHA, and PRN following chemical detoxification. This manuscript addresses an important problem in this field with helpful information. However, there are significant issues. I recommend that this manuscript be published with major revisions. The comments and suggestions are listed below.
1. Line 20: It is well known that the quality of polyclonal antibodies varies from batch to batch. This issue needs to be discussed.
2. Line 43: Five antigen components? The authors only listed four antigens.
3. Line 73: This work only focuses on PT, FHA, and PRN. The authors should also discuss dealing with the other antigen, Fim.
4. Lines 15, 20, 44, 73, and others: Initially, PT toxin (PTx) and PT toxoid (PTd) were used. Yet, in many places, the authors used PT. Please be consistent and specific.
5. Line 266: Table 2: Results on validation of precision? Please correct it.
6. Table 2: Thermotical value? Or theoretical value? If they are theoretical values, what are these values? How are these values chosen? Do they cover the low and high levels (25 ug/ml to 200ug/ml)? It is critical to evaluate the accuracy and precision using all levels of detection limits.
7. Line 276: Results on validation of accuracy? Please correct it.
8. Line 287: ….in Materials and Methods section (). Please fill in the blank.
Author Response
Comment 1: Line 20: It is well known that the quality of polyclonal antibodies varies from batch to batch. This issue needs to be discussed.
Response 1: Thank you for pointing this out; we completely agree. To ensure the quality of each batch sera, we performed a double immunodiffusion test to determine the titer, selecting only those with a titer greater than 8. In addition, we plan to examine the effects of different batches of rabbit sera (polyclonal antibodies) on the robustness of the assay in our upcoming schedule, which we have discussed this point in the revised manuscript- Page 10, Lines 359-361 [Notably, the quality of rabbit sera varies between batches, and this variation's effect on the assay's robustness should be investigated in future work.].
Comment 2: Line 43: Five antigen components? The authors only listed four antigens.
Response 2: Thanks for pointing this out. The missing information is that fimbrial agglutinogens (Fim) consist of Fim 2 and Fim 3. We have corrected this mistake and highlighted it in the revised manuscript-Page 1, Line 43 [ fimbrial agglutinogens (Fim 2/3)].
Comment 3: Line 73: This work only focuses on PT, FHA, and PRN. The authors should also discuss dealing with the other antigen, Fim.
Response 3:That is a good point that we should have discussed. According to the work of Dr. Xing and colleagues, this assay was not suitable for them; in addition, we didn’t include the two Fim antigens in our current vaccine formulations. We have added the discussion on this point in the revised manuscript-Page 10, Lines 360-363 [ In addition, according to Dr. Xing and colleagues' work, the assay was unsuitable for the other two components, Fim2 and Fim3. Since these components were not included in our vaccine formulations, we did not test Fim2 and Fim3 in the SRD system.].
Comment 4: Lines 15, 20, 44, 73, and others: Initially, PT toxin (PTx) and PT toxoid (PTd) were used. Yet, in many places, the authors used PT. Please be consistent and specific.
Response 4: We apologize for any inconsistencies that may have affected your reading experience. We have thoroughly reviewed the entire manuscript for these issues and made the necessary corrections. You can find the corrections in the updated manuscript-Lines 73,94,109,111,119,149,159,169,-178,181,191,227(Table 1),230,231,252,257,259-261,265,269(Table 2),271,274,297,300,330,367,381.
Comment 5: Line 266: Table 2: Results on validation of precision? Please correct it.
Response 5: We apologize for the typographical error and have corrected it in the revised manuscript-Page 7, Line 269 [Table 2. Results on validation of accuracy], and we have reviewed the entire manuscript for any other typographical errors.
Comment 6: Table 2: Thermotical value? Or theoretical value? If they are theoretical values, what are these values? How are these values chosen? Do they cover the low and high levels (25 ug/ml to 200ug/ml)? It is critical to evaluate the accuracy and precision using all levels of detection limits.
Response 6: We apologize for using the incorrect word, which should be "theoretical value." Unfortunately, the values provided did not cover the lower level. We greatly appreciate your valuable feedback on this matter and will seriously consider it as we conduct more thorough and robust validation of accuracy and precision in the future. Additionally, we have discussed this point in the revised manuscript-Page 10, lines 356-358 [ It is important to note that the theoretical values of the selected spiked samples did not cover the lower limit of the detection range. In other words, more thorough validation of accuracy and precision should be conducted in the future ].
In addition, we would like to explain and clarify how we designed the experiments for validating accuracy and precision. These experiments were conducted in accordance with the instructions outlined in section 3.3 of the "ICH Q2(R2) Guideline - Validation of Analytical Procedures" and the "Chinese Pharmacopeia - 9101 Guidelines for Validation of Analytical Methods".
To validate accuracy, we selected three different concentrations of spiked PTd samples. To prepare these spiked samples, we mixed PTx with high, medium, and low antigen content in equal volumes with a batch of PTd. We calculated the "theoretical antigen contents" of the three spiked samples based on the amounts of PTx and PTd used. Subsequently, the spiked samples were tested three times to obtain the "measured values." Finally, we calculated and evaluated the recovery and relative standard deviation (RSD). The exact process was applied for FHA and PRN.
We investigated repeatability and intermediate precision for precision validation by performing at least six determinations at 100% of the test concentration, as described in the ICH Q2(R2).
Comment 7: Line 276: Results on validation of accuracy? Please correct it.
Response 7: We apologize for the typographical error and have corrected it in the revised manuscript-Page 8, Line 279 [Table 3. Results on validation of precision]. In addition, we have reviewed the entire manuscript for any other typographical errors.
Comment 8: Line 287: ….in Materials and Methods section (). Please fill in the blank.
Response 8: We apologize for the typographical error and have corrected it in the revised manuscript- Page 8, Lines 289-290 [Materials and Methods section. A) PTd; B) FHA; C) PRN]. In addition, we have reviewed the entire manuscript for any other typographical errors.
Reviewer 3 Report
Comments and Suggestions for Authors
Dear authors, I have read your manuscript Development and implementation of a single radial diffusion technique for quality control of acellular pertussis vaccines, and these are my comments and suggestions (please check for facts where necessary):
Line 33: Change "mobility" to "morbidity".
Lines 36-37: Is it only in infants and young children that these vaccines do not provide lifelong protection or is this generally the case? Would an adult get lifelong protection with a single dose of either of these vaccines? Consider omitting the phrase "in infants or young children".
Lines 43-49: Change to "APV consists of up to five antigen components, including pertussis toxoid (PTd), filamentous haemagglutinin (FHA), pertactin (PRN), and fimbrial agglutinogens (Fim). However, the specific composition can vary significantly between manufacturers, and the components may be prepared by co-purifying or separate purification of the harvested material. The exact amount of each antigen is measured only in the case of separate purification, before being formulated into vaccine bulks. However, it is crucial to assay each component separately in the manufacturing process of APV to ensure quality." (Start a new paragraph after this section.)
Line 50: Change to "Protein content quantification is commonly performed for assaying ..."
Lines 61-62: Change to "An APV developed by local Chinese manufacturers using a separate purification process is currently in the clinical research stage." Omit "As far as concerned". Also, I'm not sure if the sentence fits in with the rest of the paragraph or if it should be elsewhere in the manuscript.
Line 101: Change to ", and immune sera were prepared ..."
Line 141: Change to "... using the ImmuLab Image Analysis Software", adding the company and its location in brackets.
Lines 201-202: Change to "Therefore, we selected the 250 times dilution as the optimal antiserum concentration ..."
Discussion: Please highlight what your SRD test can do that ELISA cannot do.
Line 307: Change to "has had a resurgence ...".
Comments on the Quality of English LanguageWhile the quality of English is in no way poor, I believe the clarity of the text can be improved. I have given a few examples.
Author Response
Comment 1: Line 33: Change "mobility" to "morbidity".
Response 1: We are very sorry for the incorrect writing, and we have corrected in the updated manuscript-Page 1, Line 33 [ causing morbidity and mortality worldwide]. In addition, we have reviewed the entire manuscript for any other typographical errors.
Comment 2: Lines 36-37: Is it only in infants and young children that these vaccines do not provide lifelong protection or is this generally the case? Would an adult get lifelong protection with a single dose of either of these vaccines? Consider omitting the phrase "in infants or young children".
Response 2: This is a good point. Thank you for the insightful comment. We agree with you and have corrected the text in the revised manuscript-Page 1, Lines 32-33 [ Pertussis, also known as whooping cough, remains one of the most common contagious diseases causing morbidity and mortality worldwide].
Additionally, we would like to clarify. People of all ages risk getting infected by Bordetella pertussis (whooping cough). Still, infants or young children are at the most significant risk for whooping cough and have severe complications (even death cases ) from it. Additionally, vaccination with currently available vaccines (whole-cell pertussis vaccine and acellular pertussis vaccine) will not protect life-long protection for infants, young children, and adults; in this context, the CDC recommends whooping cough vaccination for people of all ages.
Comment 3: Lines 43-49: Change to "APV consists of up to five antigen components, including pertussis toxoid (PTd), filamentous haemagglutinin (FHA), pertactin (PRN), and fimbrial agglutinogens (Fim). However, the specific composition can vary significantly between manufacturers, and the components may be prepared by co-purifying or separate purification of the harvested material. The exact amount of each antigen is measured only in the case of separate purification before being formulated into vaccine bulks. However, it is crucial to assay each component separately in the manufacturing process of APV to ensure quality." (Start a new paragraph after this section.)
Response 3: We sincerely appreciate your kind help in rephrasing the sentence. We have corrected the text in the revised manuscript- Pages 1-2, Lines 42-48. Moreover, we have decided to start a new paragraph after this section. In addition, we have reviewed the whole manuscript for correctness, clarity, engagement, and delivery and made revisions or rephrased sentences-Lines37,66-74,86,94-99,155,159,202,212-213,297,314-316,323,377-379,383-385.
Comment 4: Line 50: Change to "Protein content quantification is commonly performed for assaying ...".
Response 4: Thank you once again. We have rephrased the sentence in the updated manuscript-Page 2, Line 49 [ Protein content quantification is commonly performed for assaying...]. In addition, we have reviewed the whole manuscript for correctness, clarity, engagement, and delivery and made revisions or rephrased sentences-Lines37,66-74,86,94-99,155,159,202,212-213,297,314-316,323,377-379,383-385.
Comment 5: Lines 61-62: Change to "An APV developed by local Chinese manufacturers using a separate purification process is currently in the clinical research stage." Omit "As far as concerned". Also, I'm not sure if the sentence fits in with the rest of the paragraph or if it should be elsewhere in the manuscript.
Response 5: Thank you once again; we have rephrased the sentence in the updated manuscript-Page 2, Lines 60-61 [ An APV developed by local Chinese manufacturers using a separate purification process is currently in the clinical research stage]. In addition, we have reviewed the whole manuscript for correctness, clarity, engagement, and delivery and made revisions or rephrased sentences-Lines37,66-74,86,94-99,155,159,202,212-213,297,314-316,323,377-379,383-385.
We would like to clarify that this sentence emphasizes the importance and urgency of developing an appropriate antigen quantification assay for quality control of APV.
Comment 6: Line 101: Change to ", and immune sera were prepared ...".
Response 6: Thank you once again; we have rephrased the sentence in the updated manuscript-Page 3, Line 102 [ , and immune sera were prepared…]. In addition, we have reviewed the whole manuscript for correctness, clarity, engagement, and delivery and made revisions or rephrased sentences-Lines 37,66-74,86,94-99,155,159,202,212-213,297,314-316,323,377-379,383-385.
Comment 7: Line 141: Change to "... using the ImmuLab Image Analysis Software", adding the company and its location in brackets.
Response 7: Thank you once again; we have rephrased the sentence and added the company’s information in the updated manuscript-Pages 3-4, Lines141-143 [using the Immulab Image Analysis Software (MICROVISION Instruments SAS, CE 1750 – Z.I. Petite Montagne Sud 1 rue du Gévaudan 91047 EVRY Cedex - France)].
Comment 8: Lines 201-202: Change to "Therefore, we selected the 250 times dilution as the optimal antiserum concentration ...".
Response 8: Thank you once again; we have rephrased the sentence in the revised manuscript-Page 5, Lines 205-206 [Therefore, we selected the 250 times dilution as the optimal antiserum concentration ...].
Comment 9: Discussion: Please highlight what your SRD test can do that ELISA cannot do.
Response 9: Thank you for pointing out this; we have discussed this point in the updated manuscript-Page10, Lines 336-343 [ Glutaraldehyde, a chemical agent commonly used to treat pertussis antigens, such as PTx, FHA, and PRN, often leads to the formation of high molecular weight by-products due to its crosslinking effects. This crosslinking can obscure critical binding sites on the surface of the antigens, rendering the ELISA assay ineffective. This limitation of the ELISA assay is an important consideration in the field. In contrast, the SRD assay can accurately measure the total antigen content, regardless of any changes to the binding sites. This is accomplished using a detergent, which enables the production of diffusible subunits, irrespective of their original state].
Comment 10: Line 307: Change to "has had a resurgence ...".
Response 10: Again, thank you; we have rephrased the sentence in the revised manuscript- Page 9, Line 309 [has had a resurgence…]. In addition, we have reviewed the whole manuscript for correctness, clarity, engagement, and delivery and made revisions or rephrased sentences-Lines 37,66-74,86,94-99,155,159,202,212-213,297,314-316,323,377-379,383-385.
Round 2
Reviewer 2 Report
Comments and Suggestions for Authors
The authors addressed all my comments properly. The revised version is significantly improved.